# Sulfane Sulfur Is a Strong Inducer of the Multiple Antibiotic Resistance Regulator MarR in *Escherichia coli*

**DOI:** 10.3390/antiox10111778

**Published:** 2021-11-06

**Authors:** Huangwei Xu, Guanhua Xuan, Huaiwei Liu, Yongzhen Xia, Luying Xun

**Affiliations:** 1State Key Laboratory of Microbial Technology, Shandong University, 72 Binhai Road, Qingdao 266237, China; huangweixu@mail.sdu.edu.cn (H.X.); xuanguanhua@ouc.edu.cn (G.X.); liuhuaiwei@sdu.edu.cn (H.L.); 2School of Molecular Biosciences, Washington State University, Pullman, WA 991647520, USA

**Keywords:** sulfane sulfur, signaling, MarR, DNA binding, antibiotic resistance

## Abstract

Sulfane sulfur, including persulfide and polysulfide, is produced from the metabolism of sulfur-containing organic compounds or from sulfide oxidation. It is a normal cellular component, participating in signaling. In bacteria, it modifies gene regulators to activate the expression of genes involved in sulfur metabolism. However, to determine whether sulfane sulfur is a common signal in bacteria, additional evidence is required. The ubiquitous multiple antibiotic resistance regulator (MarR) family of regulators controls the expression of numerous genes, but the intrinsic inducers are often elusive. Recently, two MarR family members, *Pseudomonas aeruginosa* MexR and *Staphylococcus aureus* MgrA, have been reported to sense sulfane sulfur. Here, we report that *Escherichia coli* MarR, the prototypical member of the family, also senses sulfane sulfur to form one or two disulfide or trisulfide bonds between two dimers. Although the tetramer with two disulfide bonds does not bind to its target DNA, our results suggest that the tetramer with one disulfide bond does bind to its target DNA, with reduced affinity. An MarR-repressed mKate reporter is strongly induced by polysulfide in *E. coli*. Further investigation is needed to determine whether sulfane sulfur is a common signal of the family members, but three members sense cellular sulfane sulfur to turn on antibiotic resistance genes. The findings offer additional support for a general signaling role of sulfane sulfur in bacteria.

## 1. Introduction

Sulfane sulfur is a common cellular component, and it exists in several forms, including hydrogen polysulfide (H_2_S_n_, *n* ≥ 2), organic polysulfide (RSS_n_H, RSS_n_R, *n* ≥ 2), and elemental sulfur (S_8_) [1,2]. Cellular sulfane sulfur is produced from sulfide (H_2_S and HS^−^) oxidation by quinone oxidoreductases and flavocytochrome *c* sulfide dehydrogenases [3,4], or from the metabolism of l-cysteine by 3-mercaptopyruvate sulfurtransferase and cysteinyl-tRNA synthetase [5,6,7]. Excessive sulfane sulfur is removed either by enzymes, including persulfide dioxygenase [8], thioredoxin, and glutaredoxin [9], or by chemical reduction to H_2_S with cellular thiols like glutathione [10]. Although H_2_S has been recognized as a potential signaling molecule, recent evidence suggests that H_2_S signaling takes place when sensing proteins detect sulfane sulfur [11,12]. Cellular sulfane sulfur in bacteria is maintained in a micromolar range, changing with growth phases and likely regulating biological behaviors associated with growth phases [2,12].

The signaling role of H_2_S and sulfane sulfur has been studied extensively in mammalian cells [13,14], but limited research has been conducted on microorganisms. Initially, the signaling function of sulfane sulfur is identified with gene regulators, such as FisR [15], CstR [16], SqrR [17], and CsoR [18], which regulate sulfur metabolism. Recently, other gene regulators that sense sulfane sulfur have been reported. The bacterial gene regulator OxyR, which responds to oxidative stress, also senses high levels of cellular sulfane sulfur to activate the expression of thioredoxin, glutaredoxin, and catalase, which remove sulfane sulfur [19]. The activity of the quorum-sensing regulator LasR is enhanced by sulfane sulfur, which reaches its maximal level at the early stationary phase in *Pseudomonas aeruginosa* PAO1 [20]. Sulfane sulfur also deactivates MexR to express an efflux pump (MexAB) for antibiotic resistance when *P. aeruginosa* PAO1 enters the stationary phase [12]. MgrA is modified by sulfane sulfur when *Staphylococcus aureus* is under H_2_S stress to derepress genes involved in antibiotic resistance and virulence [21]. Sulfane sulfur modifies the Cys residues in the above regulators, producing persulfide (RSSH), trisulfide (RSSSH), disulfide (RSSR), or multiple sulfur links (RS_n_R, *n* ≥ 3).

Both MexR and MgrA belong to the MarR (multiple antibiotic resistance regulator) family. MarR family transcriptional regulators are ubiquitous in bacteria, modulating numerous cellular processes, and they are well known to sense and exert resistance against multiple antibiotics, detergents, and oxidative reagents [22,23,24]. The *E. coli* MarR is the prototypical member of the MarR family [25]. MarR mutations allow *E. coli* to develop resistance to multiple antibiotics [26], oxidative stress agents [27], and organic solvents [28]. MarR represses the expression of the *marRAB* operon, encoding itself (MarR), the global gene regulator MarA, and the unknown function protein MarB. MarA activates the expression of genes for resistance to multiple antibiotics [29]. Salicylate at high concentrations is able to fully induce the expression of the *marRAB* operon [25], but the high concentration makes it physiologically irrelevant. Recently, Cu^2+^ has been observed to catalyze the formation of two disulfide bonds between four Cys80 residues of two MarR dimers, leading to the formation of a tetramer with disrupted DNA binding [30]. Free Cu^2+^ is proposed to be released from copper-containing proteins upon exposure to multiple antibiotics.

Since sulfane sulfur is sensed by two MarR family regulators, MexR and MgrA [12,21], we tested whether sulfane sulfur modified MarR. As expected, inorganic polysulfide (HS_n_H, *n* ≥ 2) readily induced the formation of the disulfide bond between Cys80 residues to produce MarR tetramers. HS_n_H modification reduced MarR’s affinity to its target DNA. An MarR reporter system showed that MarR responded to exogenously added HS_n_H and salicylate.

## 2. Materials and Methods

### 2.1. Bacterial Strains, Culture Conditions, and Reagents

Strains and plasmids used in this work are listed in Table 1. *E. coli* BL21(DE3) was grown in LB medium at 37 °C with shaking. Kanamycin (50 μg/mL) and gentamicin (30 μg/mL) were added when required. All the primers used in this study are listed in Table 2. The sulfide (NaHS, H_2_S donor) was purchased from Sigma-Aldrich (St. Louis, MO, USA).

### 2.2. H_2_S_n_ Preparation

H_2_S_n_ was prepared according to a previous report [12]. Briefly, 25.6 mg of sulfur powder, 32 mg of NaOH, and 44.8 mg of NaHS were added to 20 mL of anoxic distilled water under argon gas. The bottle was sealed and incubated at 37 °C until sulfur powder was completely dissolved. A cyanolysis method was used to determine H_2_S_n_ concentration, with thiosulfate as the standard for calibration [31].

### 2.3. Reporter Construction and Test

The reporter plasmid pBBR5-MarR-P*_marR_*-mKate was constructed by placing MarR under P*lac* of the vector and the *mkate* gene after the *marR* promoter (P*_marR_*) from *E. coli* in pBBR1MCS5 [25,32]. P*_marR_* had two MarR-binding sites. The LacI-binding site (the *lac* operator) of P*lac* was deleted during plasmid construction to allow constitutive expression of MarR. We used an established method for plasmid construction [33]. The constructed plasmid was transformed into *E. coli* BL21(DE3).

The reporter strains were grown at 37 °C with shaking in LB medium to OD_600nm_ of 0.6, and 600 µM H_2_S_n_, 600 µM NaHS, 600 µM H_2_O_2_, or 4 mM salicylate was added. After incubating at 37 °C for 2 h to produce mKate, the culture OD was adjusted to 1, and 0.2 mL was transferred to a 96-well plate for mKate fluorescence measurement. For this measurement, we used the SynergyH1 microplate reader. The excitation wavelength was set at 588 nm, and the emission wavelength was set at 633 nm.

### 2.4. Protein Expression and Purification

The *marR* gene was amplified and cloned into pET-30a vector between NdeI and XhoI sites. The recombinant MarR had an N-terminal His tag for purification. Cloning was conducted using the reported method [33], and site-directed mutagenesis of the cloned genes was conducted using a modified QuikChange method [34] to generate MarR-5XS, in which 5 Cys residues were changed to Ser residues. *E. coli* BL21(DE3) samples carrying the expression plasmids were cultured in LB medium at 37 °C until OD_600nm_ reached about 0.4–0.6, and then 0.5 mM isopropyl b-d-1-thiogalactopyranoside (IPTG) was added. The temperature was changed to 25 °C for overnight cultivation. Cells were harvested by centrifugation and disrupted through a high-pressure crusher, SPCH-18 (STANSTED), at 4 °C, in an ice-cold lysis buffer (50 mM NaH_2_PO_4_, 300 mM NaCl and 20 mM imidazole, pH 8.0). The sample was centrifuged and the supernatant was loaded onto the nickel–nitrilotriacetic acid (Ni-NTA) agarose resin (Invitrogen, Waltham, MA, USA). The target protein was purified following the manufacturer’s instructions. The eluted protein was loaded onto a PD-10 desalting column (GE) for buffer exchange to a 20 mM sodium phosphate buffer (pH 7.6). Protein purification was performed under anaerobic conditions and all buffers used were fully degassed. Purity of the proteins was examined via sodium dodecyl sulphate–polyacrylamide gel electrophoresis (SDS-PAGE). Non-reducing SDS-PAGE was down by omitting the reducing agent dithiothreitol (DTT) in the loading buffer.

### 2.5. Electrophoretic Mobility Shift Assay (EMSA)

A 100-bp DNA probe containing the *marRAB* promoter sequence and ending at 25 bp before the *marR* gene was obtained by using PCR from the *E. coli* MG1655 genomic DNA. The EMSA reaction mixtures were set up in a final volume of 15 μL containing different amounts of MarR or MarR-5XS, 20 nM DNA probe, and the binding buffer (10 mM Tris, 50 mM KCl, 5% glycerin, pH 8.0). After incubating at 25 °C for 30 min, the reaction mixture was loaded onto a 6% native polyacrylamide gel and electrophoresed at 180 V for 1.5 h. The gel was stained with SYBR green I and photographed with a FlourChemQ system (Alpha Innotech, San Leandro, CA, USA).

### 2.6. Size Exclusion Chromatography

Freshly purified protein of MarR-5XS (25 µM) was incubated with 400 μM H_2_S_n_ for 20 min and analyzed with size exclusion chromatography by using a Superdex-200 (GE Healthcare, Chicago, IL, USA) column equilibrated with gel filtration buffer (10 mm Tris-HCl, pH 8.0, 100 mm NaCl). The molecular mass of the eluting tetramer species was estimated with bovine serum albumin (Mw: 67 KDa) and T7 RNA polymerase (Mw: 99 KDa) as the reference.

### 2.7. LC–MS/MS Analysis

The purification of MarR used for MS analysis was performed in an anaerobic chamber containing 95% N_2_ and 5% H_2_. The protein was reacted with 10-fold (molar ratio) H_2_S_n_ under anaerobic conditions at room temperature for 30 min. The denaturing buffer (0.5 M Tris-HCl, 2.75 mM EDTA, 6 M Guanidine-HCl, pH 8.1) and excess iodoacetamide (IAM) were added to denaturalize protein and block free thiols. Samples were digested by trypsin (Promega, Madison, WI, USA) for 12 h at 37 °C and were subject to C18 Zip-Tip (Millipore, Burlington, MA, USA) purification for desalting. Following this, they were analyzed by HPLC–tandem mass spectrometry (LC–MS) through use of a Prominence nano-LC system (Shimadzu, Kyoto, Japan) and an LTQ-OrbitrapVelos Pro CID mass spectrometer (Thermo Scientific, Waltham, MA, USA). A linear gradient of solvent A (0.1% formic acid in 2% acetonitrile) and solvent B (0.1% formic acid in 98% acetonitrile) from 0% to 100% (solvent B) in 100 min was used for elution. Full-scan MS spectra (from 400 to 1800 *m*/*z*) were detected with a resolution of 60,000 at 400 *m*/*z*.

## 3. Results

### 3.1. MarR Sensed Sulfane Sulfur and Decreased Its DNA Binding Affinity

MarR from *E. coli* MG1655 was overproduced in *E. coli* BL21 with a N-terminal His-tag and subsequently purified. In EMSA, a 100 nM MarR (monomer) shifted a DNA probe containing a single MarR-binding site at 20 nM (Figure 1A) [35]. A complex band of an MarR dimer and the probe were dominant, and a complex band of two dimers was also present. When the MarR concentration increased, the bands shifted higher, suggesting that complexes with several MarR dimers were formed (Figure 1A). The findings are consistent with previous reports that MarR forms dimers, tetramers, and higher multimers with target DNA probes [25,32]. H_2_S_n_ treatment significantly decreased MarR’s affinity to its DNA probe, and a complete shift of the probe occurred with higher H_2_S_n_-treated MarR at 600 nM (Figure 1B). The complex of a dimer MarR with the probe was not detected, but a complex of the probe with multiple MarR dimers was detected (Figure 1B). When repeated with a 200 nM MarR, H_2_S_n_-treated MarR, and H_2_S-treated MarR, the dimer complex again disappeared after H_2_S_n_ treatment, but not after H_2_S treatment (Figure 1C).

MarR contains six cysteines (Cys47, Cys51, Cys54, Cys80, Cys108, and Cys111). Cys80 is responsible for Cu^2+^ sensing, and others can be mutated [30]. We constructed and purified the MarR-5XS mutant (with the exception of Cys80, the remaining five Cys residues were mutated to Ser). In EMSA, two bands corresponding to one MarR dimer and two MarR dimers that bound to the DNA probe were detected with the untreated MarR-5XS, and higher multimers were formed with increasing amounts of MarR-5XS (Figure 2A). H_2_S_n_ treatment decreased MarR-5XS’s affinity to its probe, as the unshifted probe was detectable in the presence of 100 nM H_2_S_n_-treated MarR-5XS, but not in the presence of 100 nM untreated MarR-5XS (Figure 2A). H_2_S_n_ treatment eliminated the probe complexed with an MarR-5XS dimer but enhanced the probe complexed with two dimers. The probe formed complexes with multiple dimers, with increased MarR-5XS and H_2_S_n_-treated MarR-5XS (Figure 2A). Size exclusion chromatography revealed that MarR-5XS was primarily present as a dimer in the solution, but H_2_S_n_ treatment produced tetramers (Figure 2B). The peak at around 9 min may represent multimers of MarR-5XS after H_2_S_n_-treatment (Figure 2B). On the basis of the peak areas, the dimer, tetramer, and multimer represented 37.4%, 54.1%, and 8.5% of the H_2_S_n_-treated MarR proteins, respectively. In a non-reducing SDS-PAGE gel, untreated MarR-5XS mainly produced a monomer band, and H_2_S_n_-treated MarR-5XS showed both bands of monomers and dimers (Figure 2C). On the basis of the band intensities, the monomer and dimer accounted for 33.0% and 67.0% of the H_2_S_n_-treated MarR proteins, respectively. These results suggest that the H_2_S_n_ treatment forms MarR tetramers that have reduced affinity to their probes.

H_2_S_n_-treated MarR was digested by trypsin and analyzed by using liquid chromatography–mass spectrometry (LC–MS). A 2^+^-charged peak (*m*/*z*: 461.27) matching the disulfide (Cys^80^-Cys^80^)-containing peptide (theoretical molecular mass: 921.53Da) and a 2^+^-charged peak (*m*/*z*: 477.26) matching the trisulfide (Cys^80^-S-Cys^80^)-containing peptide (theoretical molecular mass: 953.53Da) were identified from the MS data of H_2_S_n_-treated MarR (Figure 3A,B). These results indicate that H_2_S_n_ induces the formation of disulfide and trisulfide bonds between two MarR dimers via Cys^80^.

### 3.2. MarR Responded to H_2_S_n_ in An E. Coli Reporter System

We constructed a reporter plasmid, pBBR5-MarR-P*_marR_*-mKate, in which mKate is repressed by constitutively expressed MarR, and transformed it into *E. coli* BL21(DE3). The mKate expression was induced by H_2_S_n_ in a dose-dependent response (Figure 4A). We also checked mKate induction by a known inducer, salicylate, and potential inducers, H_2_S and H_2_O_2_. The mKate fluorescence was increased significantly by H_2_S_n_, salicylate, and H_2_O_2_, but not by H_2_S when added to the cell suspension (Figure 4B). Among the tested inducers, H_2_S_n_ was the strongest for inducing mKate expression.

## 4. Discussion

As illustrated in Figure 5, we demonstrated that MarR senses sulfane sulfur. After reacting with H_2_S_n_, MarR forms a disulfide bond (Cys80-Cys80′) or a trisulfide bond (Cys80-S-Cys80′) between two dimers, generating a tetramer in solution (Figure 2B). When the tetramer is formed by Cu^2+^ in the presence of O_2_, two disulfide bonds are formed between the two dimers, burying the alpha helix containing the Cys^80^ residue for DNA binding. Therefore, the structure suggests that the tetramer does not bind to its target DNA [30]. The H_2_S_n_ treatment may induce the formation of tetramers with one disulfide bond as well as two disulfide bonds (Figure 5), and the tetramer with a single disulfide bond may still bind to its DNA probe, as evidenced by the presence of the tetramer–probe complex and lack of the dimer–probe complex (Figure 1 and Figure 2A). Size exclusion chromatography showed the dominance of tetramers, but the non-reducing SDS-PAGE analysis indicated an abundance of monomers of the H_2_S_n_-treated MarR-5XS (Figure 2), further suggesting the presence of tetramers with a single disulfide bond (Figure 5). The formation of disulfide and trisulfide bonds reduced their affinities to the target DNA, derepressing the repressed genes in both EMSA and in vivo reporter assays (Figure 1 and Figure 4). These results are similar to those regarding the response of *P. aeruginosa* MexR to sulfane sulfur, forming interprotomer disulfide and trisulfide bonds that covalently link the monomers within a dimer that fails to bind to its cognate DNA [12]. The cellular level of sulfane sulfur varies according to the growth phases of bacteria in liquid media [2]. *P. aeruginosa* contains the highest level of sulfane sulfur in the early stationary phase, which inactivates MexR for the expression of drug resistant genes [12]. MarR may also sense cellular levels of sulfane sulfur to regulate its repressed genes.

MarR family proteins are small proteins with a winged helix–turn–helix (wHTH) DNA-binding domain and the *N*- and *C*-termini participating in the dimer formation [23]. The MarR family proteins are common in bacteria, averaging seven different genes per sequenced bacterial genome [9]. Some are involved in the regulation of efflux pumps for the resistance of antibiotics and organic solvents [23,30,36,37]. Other members display diverse regulatory functions. For example, MgrA is involved in the regulation of biofilm formation and virulence in *S. aureus* [38,39]. OhrR regulates resistance to organic hydroperoxides in *Pseudomonas aeruginosa* [40]. HcaR regulates the utilization of hydroxycinnamate in *Acinetobacter* sp. ADP1 [41]. SlyA and homologues in Enterobacteriaceae have evolved to facilitate the expression of horizontally acquired genes [42]. Their mechanisms of sensing are either binding salicylate or other aromatic compounds in a crevice between the DNA binding domain and the dimer interface [41], or using Cys thiols to detect oxidative stress [30]. Some members, such as MarR, use both mechanisms for inducer sensing (Figure 4B).

Three MarR family proteins that use Cys thiols to sense oxidative stress also sense sulfane sulfur. *E. coli* MarR is oxidized by Cu^2+^ to form two disulfide bonds between two dimers (Cys80-Cys80′), turning on the expression of the *marRAB* operon for further activation of other genes, including a multiple drug efflux pump for antibiotic resistance [30]. When exposed to antibiotics, Cu^2+^ is released from membrane proteins under oxidative stress. *P. aeruginosa* MexR responds to H_2_O_2_ as well as disulfides GSSG and 2,2′-dithiodipyridine to form interprotomer disulfide bonds between Cys^30^ and Cys^62^ in an MexR dimer [36,37]. *S. aureus* MgrA contains a single Cys residue, Cys12, which is modified by hydrogen peroxide, and the modified MgrA has reduced affinity to its target DNA [43]. These MarR family proteins, which use Cys thiols for sensing, can be deactivated by sulfane sulfur to derepress their regulated genes [12,21]. Further studies are necessary to determine which other members of the MarR family can sense sulfane sulfur.

Sulfane sulfur is the inducer of several gene regulators involved in sulfur metabolism [44]. After reacting with sulfane sulfur, *Cupriavidus pinatubonensis* FisR forms a tetrasulfide, crosslinking between C53 and C64 [15]. *S. aureus* CstR generates a mixture of di-, tri-, and tetra-sulfide links between C31 and C60 [16]. *Rhodobacter capsulatus* SqrR produces a tetrasulfide link between C41 and C107 [17]. These modifications may be induced specifically by sulfane sulfur, as other oxidative reagents, such as H_2_O_2_, organic peroxides, and disulfide, cannot induce the formation of multiple sulfur links between two Cys residues. This concept has been applied to design sulfane sulfur-specific fluorescence proteins which, due to the space restriction of the Cys residues involved, cannot form disulfide links, but can form trisulfide and tetrasulfide links [45,46]. Disulfide formation can be achieved by the induction of sulfane sulfur as well as H_2_O_2_ and other oxidative reagents. Previously, Cu^2+^ induced the formation of a disulfide bond by MarR, but H_2_O_2_ did not [30]. However, we detected H_2_O_2_ as a reducer for MarR (Figure 4B). Given the chemistry of disulfide bond formation, H_2_O_2_, H_2_S_n_, and Cu^2+^ should all induce the formation of the MarR disulfide bond (Figure 5).

## 5. Conclusions

Sulfane sulfur is the specific inducer that modified gene regulators involved in sulfur metabolism [44]. It is also an effector of OxyR [19], MexR [12], MgrA [21], and MarR. Furthermore, it may sense other agents, such as H_2_O_2_, as long as they can induce disulfide formation or modify key Cys thiols. MarR regulates diverse activities in *E. coli*, including resistance to antibiotics, oxidative stress agents, and organic solvents. Although several inducers have been reported, sulfane sulfur, a common cellular component in bacteria that reaches the maximum in *E. coli* cells at the late log phase and the early stationary phases of growth [2], is likely a key inducer regulating MarR activity. This discovery further establishes the general signaling role of sulfane sulfur.

## Figures and Tables

**Figure 1 antioxidants-10-01778-f001:**
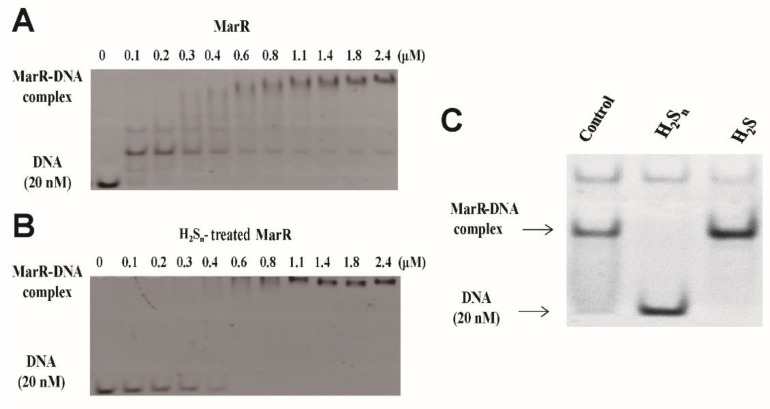
MarR reacts with sulfane sulfur, decreasing its affinity to its DNA probe. (**A**,**B**) EMSA analysis of MarR and H_2_S_n_-treated MarR in the presence of 20 nM the DNA probe (100 bp) containing the MarR-binding sequence. (**C**) The purified MarR (9 µM) was treated with 800 µM H_2_S_n_ and H_2_S for 20 min. Then 20 nM DNA probe was incubated with 0.2 µM MarR for the shift assay.

**Figure 2 antioxidants-10-01778-f002:**
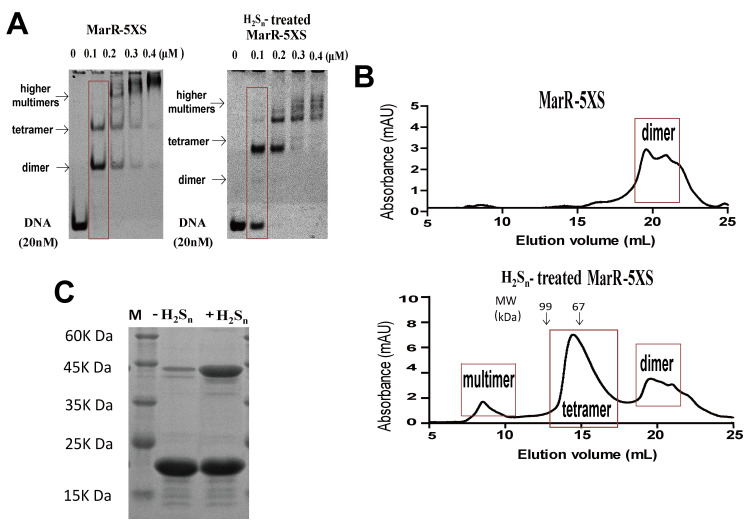
MarR sensed sulfane sulfur via Cys80. (**A**) EMSA analysis of H_2_S_n_-treated and untreated MarR-5XS in the presence of its DNA probe (20 nM). (**B**) Size-exclusion chromatography revealed the formation of disulfide-stabilized tetramer of MarR-5XS (25 µM) after 400 μM H_2_S_n_ treatment for 20 min. The molecular weight (MW) of tetramers was calibrated with BSA (67 kDa) and T7 RNA polymerase (99 kDa). (**C**) Non-reducing SDS-PAGE analysis of MarR-5XS (25 µM). The protein was treated with H_2_S_n_ (400 μM) for 20 min, terminated by adding 500 μM iodoacetamide before analysis.

**Figure 3 antioxidants-10-01778-f003:**
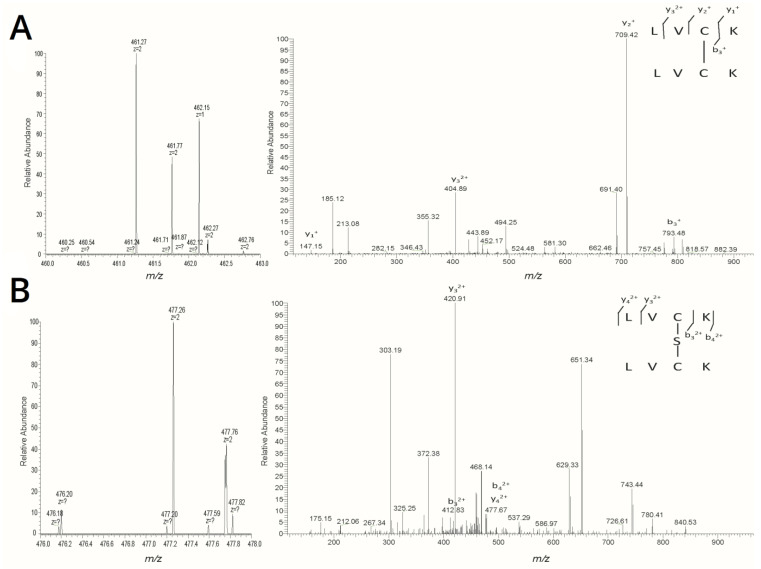
LC–MS/MS analysis of H_2_S_n_-treated MarR. (**A**) Left: The 2^+^ charged peak (*m*/*z*: 461.27) corresponding to the disulfide (Cys^80^-Cys^80^)-containing peptide (theoretical molecular mass: 921.53Da). Right: MS/MS fragmentation of the 2^+^ charged peptide (*m*/*z*: 461.27). (**B**) Left: The 2^+^ charged peak (*m*/*z*: 477.26) corresponding to the trisulfide (Cys^80^-S-Cys^80^)-containing peptide (theoretical molecular mass: 953.53 Da). Right: MS/MS fragmentation of the 2^+^ charged peptide (*m*/*z*: 477.26). Peptide mass was calculated on the following website: http://db.systemsbiology.net:8080/proteomicsToolkit/FragIonServlet.html (accessed on 11 October 2021).

**Figure 4 antioxidants-10-01778-f004:**
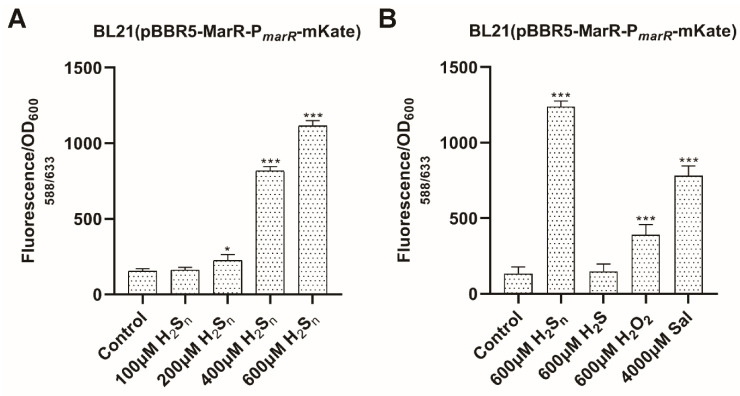
Testing MarR inducers with an *E. coli* reporter system. (**A**) A dose-dependent induction of mKate. (**B**) mKate induction by several tested inducers. *E. coli* BL21(DE3) containing pBBR5-MarR-P*_mar_**_R_*-mKate grew in LB medium to OD_600nm_ of 0.6, inducers were added, and the cultures were incubated at 37 °C for 2 h before mKate fluorescence analysis. The control was *E. coli* BL21(DE3) with the empty vector pBBR1MCS5. Student’s *t*-test was used to calculate the *p*-value. Symbol * indicates the sample is different (0.05 < *p* < 0.01) and symbol *** indicates the sample is significantly different from the control (*p* < 0.001).

**Figure 5 antioxidants-10-01778-f005:**
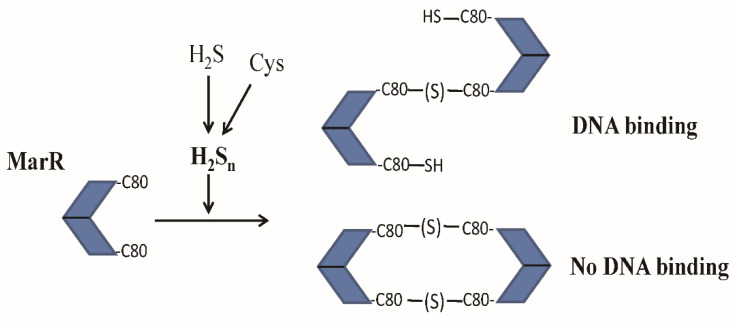
Schematic representation of MarR, which senses sulfane sulfur, forming disulfide and trisulfide crosslinks between Cys^80^. A mixture of tetramer with one and two disulfide bonds may coexist after H_2_S_n_ treatment. The tetramer with two disulfide bonds does not bind to its target DNA [30], but our results suggest that the tetramer with one disulfide bond does bind to its target DNA.

**Table 1 antioxidants-10-01778-t001:** Strains and plasmids used in this study.

Strain/Plasmid	Characteristic	Source
***Escherichia coli* strains**		
DH5_a_	Cloning strain	Invitrogen
BL21(DE3)	Cloning strain	Invitrogen
** *Plasmids* **		
pBBR1mcs5	Gmr, broad host range	[12]
pBBR5-MarR-P*_marR_*-mKate	pBBR1mcs5 vector with *marR*, *marR* promotor, and *mkate* genes	This study
pET30a	Kmr, expression vector	Invitrogen
pET30-MarR	pET30a containing MarR with N terminal his-tag	This study
pET30-MarR/5CS	pET30-MarR with Cys47Ser, Cys51Ser, Cys54Ser, Cys108Ser, Cys111Ser	This study

**Table 2 antioxidants-10-01778-t002:** Primers used in this study.

Primers ^a^	Sequence (5′-3′)	Usage
marR-1	CGACGACGACAAGGCCATGGCTGATGTGAAAAGTACCAGCGATCTG	For pET30-MarR construction
marR-2	CAGTGGTGGTGGTGGTGGTGCTCGATTACGGCAGGACTTTCTTAAG	
MarR-5CS-1	TAATACTCGCCGCGCTGCGGATAGAGCTGAGC	For MarR-5XS mutant construction
MarR-5CS-2	GCTCAGCTCTATCCGCAGCGCGGCGAGTATTA	
MarR-5CS-3	GGCGCGGCAATAAGTGAACAAAGCCAT	
MarR-5CS-4	ATGGCTTTGTTCACTTATTGCCGCGCC	
marR-F ^a^	TCTAGAGAAAGAGGAGAAATACTAGGTGAAAAGTACCAGCGATCTGTTCAAT	For pBBR5-MarR-P_*marR*_-mKate construction
marR-R	TTGACGGTGGTATTACGGCAGGACTTTCTTAAGCAAATAC	
P_mar_-F	CCTGCCGTAATACCACCGTCAAAAAAAACGGCGCTTTTTAGCGCCGTTTTTATTTTTCATGAACCGATTTAGCAAAACGTGGC	
P_mar_-R	CTAGTATTTCTCCTCTTTCTCTAGAATTAGTTGCCCTGGCAAGTAATTAGTT	
mKate-F ^b^	TCTAGAGAAAGAGGAGAAATACTAGATGTCAGAATTAATTAAAGAAAATATGCACATG	
mKate-R	CTTACAATTTCCATTCGCCATTTCAACGATGTCCTAATTTCGACG	
pBBR5-F ^c^	CTAGTATTTCTCCTCTTTCTCTAGACAACATACGAGCCGGAAGCATAAAG	
pBBR5-R ^c^	AATGGCGAATGGAAATTGTAAGCG	
MarR-emsa-1	CATCGCATTGAACAAAACTTGAAC	For amplifying the EMSA probe
MarR-emsa-1	GTTGCAGGGGATAATATTGCC	

^a^ The gene accession numbers of *marR* and *mKate* are b1530 and MN623117, respectively. The EMSA probe contained one MarR-binding site, ending at 25 bp before the *marR* gene. ^b^ An rbs sequence (gaggaga) was inserted before the start codon (GTG or ATG) of *marR* and *mkate*. ^c^ The *lac* operator sequence (ttgtgagcggataacaa) was not amplified with this primer pair.

## Data Availability

Data are contained within the article.

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
