# Peer review of "Sulfane Sulfur Is a Strong Inducer of the Multiple Antibiotic Resistance Regulator MarR in Escherichia coli"

_antioxidants, 2021, doi:10.3390/antiox10111778_

Round 1

Reviewer 1 Report

The manuscript was well written and the results obtained are interesting. A suggestion for improvement is to discuss how widespread the use of sulfane compounds as gene regulators is within bacteria generally. Unless the authors are overlooking something this is just the third bacterial species that has been shown to use sulfane sulfur to regulate gene expression. These same authors were involved in the prior studies, that now show MarR, MexR and MgrA regulatory proteins react with sulfane sulfur and it would be appropriate in this publication to expand the discussion to include the known distribution of MarR, MexR, and MgrA proteins in bacterial species, as well as the likilihood of finding other bacterial regulatory proteins that react with sulfane sulfur.

Line 157, delete "completed".

Line 287, correct spelling "althogh".

Author Response

Thank you for your suggestion on expanding the discussion about MarR family proteins. A paragraph on the topic is added. We also appreciate your revision comments, and the errors are corrected.

Reviewer 2 Report

The manuscript entitled Sulfane sulfur is a strong inducer for the multiple antibiotic resistance regulator MarR in Escherichia coli, is well written, the methods are adequately described and the results are presented in good form and clearly. Below are minor comments.

The abstract must be improved by describing the methods and results.

I recommend not including figure 5. Since the figure is not part of the data generated as results. In that case, develop the discussion in depth. 

After attending to the comments I recommend accepting the manuscript.

Author Response

Thank you for your suggestion about the abstract. We added the main results and the related methods. Your comments on Figure 5 alert us to examine the figure. We realized that our findings are not clearly presented in the figure. We revised the figure to show that some tetramers linked by a single disulfide bond and indicated that this form still binds to its cognate DNA. We also indicated the tetramer with two disulfide bonds that does not bind to its cognate DNA with a reference cited in the figure legend. The modification clear gives credits to the previous publication and also presented our findings.